# Fast and Robust Mesh Simplification for Generated and Real-World 3D Assets

Kunal Bhosikar[1]    Preet Savalia[2]    Lokender Tiwari[3]    Brojeshwar Bhowmick[3]

[1]IIIT Hyderabad    [2]IIT Jodhpur    [3]TCS Research

kunal.bhosikar@research.iiit.ac.in, b22ai036@iitj.ac.in

lokender.work@gmail.com, b.bhowmick@tcs.com

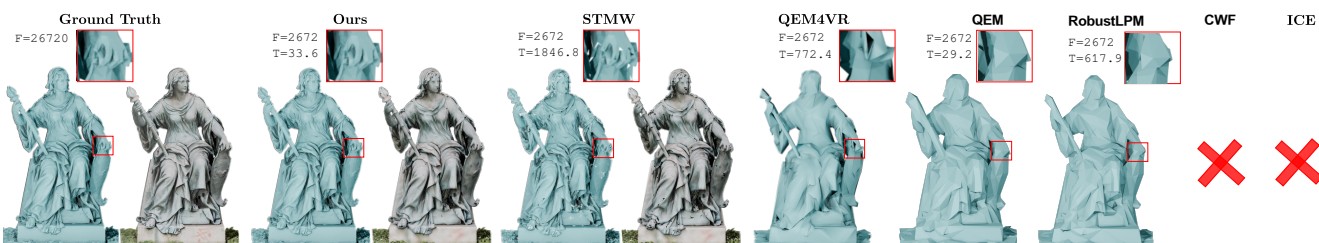

Figure 1. **FA-QEM delivers high-fidelity and efficient mesh simplification.** We compare against Liu et al. (STMW) [17], QEM4VR [1], QEM [5], RobustLPM [2], CWF [30], and ICE [16] at 10% resolution. $F$ and $T$ denote face count and runtime (s). Close-ups show that FA-QEM preserves sharp geometry and fine textures under aggressive simplification while achieving significantly lower runtimes. Several baselines do not support texture transfer. Additional results are in the supplementary.

## Abstract

*The rapid growth of 3D content from modern reconstruction and generative pipelines, such as neural rendering and large-scale 3D asset generation, has led to an abundance of dense, noisy, and often non-manifold meshes. While these representations achieve high visual fidelity, their complexity poses significant challenges for downstream applications in simulation, AR/VR, and scientific computing, where efficient and reliable geometry is essential. This necessitates mesh simplification methods that are not only fast and robust to "in-the-wild" inputs, but also capable of preserving fine geometric structures and high-quality appearance. In this paper, we propose Feature-Aware Quadric Error Metric (FA-QEM), a comprehensive mesh simplification pipeline designed for modern 3D assets. Our approach introduces a novel multi-term quadric error formulation that jointly encodes geometric deviation, boundary curvature, and surface normal consistency, enabling optimal vertex placement that preserves sharp features even under aggressive simplification. Furthermore, we show that high-fidelity geometric simplification significantly improves downstream appearance transfer, serving as a superior front-end for texture mapping via successive mapping techniques. We conduct extensive evaluations on both AI-generated meshes and large-scale real-world datasets, including Thingi10K and the Real-World Textured Things dataset. Our results demonstrate that FA-QEM achieves consistently lower geometric error, better visual fidelity, and substantially faster runtimes compared to existing methods, while maintaining robustness across diverse and challenging inputs. These properties make FA-QEM a practical and effective component for scalable 3D reconstruction and generation pipelines.*

## 1. Introduction

Recent advances in 3D reconstruction and generative models, such as Neural Radiance Fields (NeRF) [22] and 3D Gaussian Splatting (3DGS) [13], have led to a rapid increase in high-fidelity 3D assets. However, meshes extracted from these pipelines [6, 18, 23] are often dense, noisy, and non-manifold, limiting their usability in downstream applications such as simulation, AR/VR, and scientific computing.

To make these assets practical for downstream tasks such as simulation, scientific computing, and VR/AR, they must be converted into efficient geometric representations. However, a trade-off exists between visual fidelity and computational efficiency. High-polygon meshes impose significant computational overload, leading to reduced performance, higher latency, and inefficiencies in downstream pipelines.

Mesh simplification is a key step in addressing this challenge, reducing geometric complexity while preserving visual appearance. The Quadric Error Metric (QEM) [5] re-

mains a foundational approach due to its efficiency and effectiveness. However, as 3D data has become more diverse and complex, traditional QEM [5] and its variants [4] struggle to handle modern "in-the-wild" meshes arising from reconstruction and generative pipelines. Robust methods [17] capable of handling non-manifold and noisy inputs often sacrifice fine geometric details, while feature-preserving approaches [1] tend to be brittle and restrict achievable simplification. This creates a fundamental trade-off between robustness, fidelity, and efficiency.

In addition, computational efficiency remains a major bottleneck. Many high-quality simplification techniques are too slow for practical deployment in large-scale 3D pipelines, limiting their applicability in real-world systems such as simulation environments, digital twins, and interactive applications. Consequently, there is a critical need for mesh simplification methods that are simultaneously fast, robust to real-world inputs, and capable of preserving both geometric and appearance fidelity.

In this paper, we introduce **FA-QEM** (**F**eature-**A**ware **Q**uadric **E**rror **M**etric), a comprehensive mesh simplification pipeline designed for modern 3D assets. Our method addresses the trilemma of speed, robustness, and fidelity through a novel multi-term QEM formulation that incorporates curvature-aware boundary constraints and normal preservation to maintain sharp geometric features under aggressive simplification. Furthermore, we show that high-quality geometric simplification serves as a critical front-end for appearance transfer, significantly improving the effectiveness of texture mapping via successive mapping techniques.

Our contributions are: (1) A unified multi-term QEM formulation for joint geometry and feature preservation, (2) A geometry-first pipeline that improves downstream texture transfer, (3) A fast and robust implementation for in-the-wild meshes.

Through extensive qualitative and quantitative evaluations, we show that FA-QEM produces simplified meshes with superior geometric accuracy, visual fidelity, and runtime performance across diverse datasets, including both AI-generated and real-world meshes. As illustrated in Fig. 1, our approach consistently outperforms prior methods, enabling efficient and reliable processing of modern 3D assets. Unlike prior QEM-based extensions that treat constraints independently, FA-QEM provides a unified multi-objective formulation that balances robustness, fidelity, and efficiency in a single optimization framework.

## 2. Related Work

**Foundational Work.** Mesh simplification is a core topic in graphics and 3D vision [3, 20]. Early work on remeshing [11, 28], clustering [19, 24], and decimation [7, 25] led to edge collapse methods [9] and the Quadric Error Metric

(QEM) [5], which remains widely used due to its efficiency and fidelity.

**Attribute-Preserving QEM.** Extensions of QEM preserve attributes such as boundaries [4], volume [15], color [4, 10], probabilistic distributions [27], and intrinsic geometric properties [16]. However, these typically treat constraints independently. In contrast, FA-QEM uses a unified multi-term quadric encoding boundary curvature, normals, and geometry, enabling more stable, feature-aware simplification for complex meshes.

**Meshes in the Wild and Texturing.** Modern reconstruction and generative methods produce dense, noisy, and often non-manifold meshes. Outputs from NeRF [22], 3DGS [13], and related approaches extracted via Marching Cubes [18] or hybrid methods [26] are particularly challenging. Liu et al. (STMW) [17] addresses robustness via successive mapping, but appearance quality depends on geometric fidelity. FA-QEM improves this by producing higher-quality geometric "canvases" for texture transfer.

**Other Strategies and Our Positioning.** Alternative methods include variational remeshing [2, 14] and feature consolidation [30], which are often computationally expensive. Unlike remeshing-based methods, FA-QEM preserves input topology, contributing to both efficiency and stability. Learning-based approaches [8] require training and struggle with out-of-distribution meshes. In contrast, FA-QEM is a general-purpose, training-free method designed as a lightweight component in modern 3D pipelines, enabling efficient conversion of dense meshes into compact, high-quality assets.

## 3. Method

Our goal is to design a fast and robust mesh simplification method tailored for modern 3D pipelines, where dense and unstructured meshes produced by reconstruction and generative models must be converted into efficient, high-fidelity geometric assets. These settings arise in applications such as scientific computing, simulation, digital twins, and immersive AR/VR, where preserving salient geometry and appearance under aggressive reduction is essential. While classic QEM [5] focuses solely on minimizing geometric deviation, it often fails on modern, complex assets that exhibit noise, non-manifold topology, and high-frequency details. We extend QEM with multi-objective penalties tailored to these challenges, yielding a more expressive error metric that preserves both geometry and appearance in real-world scenarios.

Our method **FA-QEM** is a two-stage pipeline designed as a lightweight post-processing module for modern 3D as-

sets, as illustrated in Figure 2. In the first stage, we compute optimal edge collapse positions using our proposed multi-objective QEM formulation. The core of our algorithm is an iterative edge collapse process guided by a priority queue over candidate edges. At each iteration, the lowest-cost edge is collapsed, and neighboring costs are updated until the desired target resolution is reached. In the second stage, we transfer appearance from the original mesh to the simplified mesh using a successive mapping strategy, ensuring high-fidelity texture preservation.

We discuss each of these steps in the following sections.

## 3.1. Preliminaries - Quadric Error Metric

We briefly outline the Quadric Error Metric (QEM) framework [5]. QEM associates a $4 \times 4$ symmetric matrix $Q$ with each vertex, encoding the sum of squared distances to its incident face planes. The error for a vertex $\mathbf{v} = [x, y, z, 1]^T$ is computed as $\Delta(\mathbf{v}) = \mathbf{v}^T Q \mathbf{v}$.

The quadric for a vertex $\mathbf{v}_i$ is the sum of fundamental quadrics $K_p = \mathbf{p}\mathbf{p}^T$ for all its incident planes $\mathbf{p}$ (where $\mathbf{p}$ defines the plane equation $\mathbf{p} \cdot \mathbf{x} = 0$).

$$K_p = \mathbf{p}\mathbf{p}^T, \qquad Q^i = \sum_{p \in \text{planes}(\mathbf{v}_i)} K_p. \qquad (1)$$

When collapsing an edge $(\mathbf{v}_i, \mathbf{v}_j)$ to a new vertex $\mathbf{v}'$, the new quadric is simply $Q' = Q^i + Q^j$. The optimal position for $\mathbf{v}'$ is found by minimizing the error $\Delta(\mathbf{v}') = \mathbf{v}'^T Q' \mathbf{v}'$, which reduces to solving a small $3 \times 3$ linear system. Following standard practice, if this system is singular or ill-conditioned, we fall back to selecting the vertex from the set $\{\mathbf{v}_i, \mathbf{v}_j, (\mathbf{v}_i + \mathbf{v}_j)/2\}$ that yields the minimum error $\Delta$.

## 3.2. FA-QEM: Proposed Feature-Aware QEM

Our proposed method, FA-QEM, is a two-stage pipeline for geometry optimization and appearance preservation in modern 3D assets. It is designed to robustly simplify meshes produced by reconstruction and generative pipelines while maintaining high visual fidelity. The first stage (Sec. 3.2.1) introduces our core contribution: a composite quadric formulation for feature-aware geometric simplification. The second stage (Sec. 3.2.3) performs texture transfer via successive mapping on the resulting high-quality geometry.

The geometric simplification procedure is detailed in Algorithm 1. We first initialize a composite geometric-feature quadric, $Q_{gf}$, for every vertex (Line 2). This $Q_{gf}$ is a weighted sum of three distinct components: a base quadric, a boundary/curvature quadric, and a normal quadric (detailed in Sec 3.2.1). We then populate a priority queue with all edges, ranked by a total cost function (Line 3). This $cost_{total}$ (Eq. 2) combines the geometric cost from our new quadric ($cost_{gf}$) with a separate area preservation

cost ($cost_{area}$). This second term is crucial for preserving the silhouette and rim of models with open boundaries by penalizing collapses that would distort these regions. Finally, the algorithm iteratively collapses the lowest-cost edge (Lines 5-12) until the target face count is met.

$$\text{cost}_{total}(\mathbf{v}') = \text{cost}_{gf}(\mathbf{v}') + w_{area} \cdot \text{cost}_{area}(\mathbf{v}') \quad (2)$$

---

**Algorithm 1** FA-QEM

---

**Input:** Original Mesh $M_{original}$, target face count $n_{target}$
**Parameters:** All weights $\{w\}$
**Output:** Simplified textured mesh $M_{simplified}$

1: **procedure** FA-QEM($M, n_{target}$)
2:     $\{Q_{gf}^k\} \leftarrow$ INITIALIZEGFQUADRICS($M, \{w\}$)    ▷ Sec. 3.2.1
3:     $\mathcal{P} \leftarrow$ POPULATEPRIORITYQUEUE($M, \{Q_{gf}\}$) ▷ Using ComputeCost
4:     $H \leftarrow$ EmptyList()        ▷ Collapse history for mapping
5:     **while** FACECOUNT($M$) $> n_{target}$ **and not** $\mathcal{P}$.IsEmpty() **do**
6:         (cost, $e, \mathbf{v}'$) $\leftarrow \mathcal{P}$.Pop()
7:         **if not** CAUSESFLIP($e, \mathbf{v}'$) **then**
8:             $H$.Append(info($e, \mathbf{v}'$))
9:             ($M, \{Q_{gf}\}$) $\leftarrow$ COLLAPSEEDGE($M, e, \mathbf{v}', \{Q_{gf}\}$)
10:            UPDATENEIGHBORCOSTS($\mathcal{P}$, edges adjacent to collapse)
11:         **end if**
12:     **end while**
13:     $M_{simplified} \leftarrow M$
14:     TEXTUREBAKEVIASUCCESSIVEMAP($M_{simplified}, M_{original}, H$)▷ Sec. 3.2.3
15:     **return** $M_{simplified}$
16: **end procedure**
17: **function** COMPUTECOST($e(\mathbf{v}_i, \mathbf{v}_j), \{Q_{gf}\}$)
18:     $Q'_{gf} \leftarrow Q_{gf}^i + Q_{gf}^j$
19:     $\mathbf{v}' \leftarrow$ FINDOPTIMALPOSITION($Q'_{gf}$)
20:     $\text{cost}_{gf} \leftarrow (\mathbf{v}')^T Q'_{gf} \mathbf{v}'$
21:     $\text{cost}_{area} \leftarrow (\mathbf{v}')^T$COMPUTEAREAQUADRIC($e$)$\mathbf{v}'$
22:     **return** $\text{cost}_{gf} + w_{area} \cdot \text{cost}_{area}$
23: **end function**

---

The following sections will now detail the precise formulation of each component of this cost function.

### 3.2.1. Geometric Feature Cost

We define the total geometric-feature cost to collapse an edge with endpoints $(\mathbf{v}_i, \mathbf{v}_j)$ to a new vertex $\mathbf{v}'$ as

$$\text{cost}_{gf}(\mathbf{v}') = \mathbf{v}'^T Q'_{gf} \mathbf{v}' \qquad (3)$$

where $Q'_{gf}$ is the total geometric-feature quadric, which is the sum of individual quadrics of vertices $\mathbf{v}_i$ and $\mathbf{v}_j$, i.e., $Q'_{gf} = Q_{gf}^i + Q_{gf}^j$. We propose that the individual geometric-feature quadric of $k^{th}$ vertex $Q_{gf}^k$ is a composite of three key geometric components as shown in equation 4.

$$Q_{gf}^k = Q_{base}^k + Q_{boundary}^k + Q_{normal}^k \qquad (4)$$

**Area-Weighted Base Quadric ($Q_{base}$) :** To promote aggressive simplification in low-frequency surface regions, we reduce the influence of large flat triangles using inverse-area weighting and compute the total weighted base quadric as:

$$Q_{base} = \sum_{p \in \text{planes}(\mathbf{v}_i)} \frac{K_p}{w_{plane\_area} \cdot A_p} \qquad (5)$$

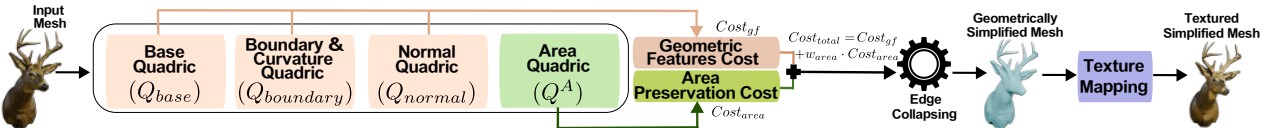

Figure 2. **Overview of the FA-QEM pipeline.** We construct a composite quadric $Q_{gf}$ from base, boundary–curvature, and normal-alignment terms to define $cost_{gf}$, alongside an area-preservation quadric $Q^A$ with cost $cost_{area}$. Edge collapses are guided by a weighted combination yielding $cost_{total}$. After simplification, successive mapping ensures consistent, high-fidelity texture transfer. See Sec. 3.2 for details.

where $K_p$ is the face-plane quadric from Eq. 1, $A_p$ is its area, and $w_{plane\_area}$ is the area weight. This formulation reduces the error penalty for collapses in flat, low-frequency regions (which have large $A_p$), thereby promoting simplification. This concentrates preservation on high-curvature, detail-rich areas (which have smaller $A_p$ and thus a higher relative cost).

**Boundary and Curvature Preservation** ($Q_{boundary}$): Sharp creases and curves are crucial geometric features of a 3D shape. To preserve such fine details, we apply an additional penalty at boundary edge vertices. A mesh edge is identified as a boundary if it belongs to only one face, and its vertices are called boundary vertices. For a boundary vertex $\mathbf{v}_1$ that has exactly two neighbors along the boundary chain, denoted $\mathbf{v}_2$ and $\mathbf{v}_3$. We estimate the local curvature $\kappa$ using a discrete approximation based on the finite differences of the boundary chains. This constraint ensures the calculation is performed on a simple, continuous boundary curve and does not incorrectly involve vertices from the interior of the mesh.

$$\kappa = \frac{\|\boldsymbol{\delta}' \times \boldsymbol{\delta}''\|}{\|\boldsymbol{\delta}'\|^3}, \quad \boldsymbol{\delta}' = \mathbf{v}_3 - \mathbf{v}_2, \quad \boldsymbol{\delta}'' = \mathbf{v}_3 - 2\mathbf{v}_1 + \mathbf{v}_2 \quad (6)$$

We then construct a curvature-scaled quadric penalty using two virtual planes. The first plane, $\mathbf{p}_1$ is orthogonal to the triangle formed by $\mathbf{v}_1, \mathbf{v}_2, \mathbf{v}_3$ and encodes the local surface orientation. The second plane, $\mathbf{p}_2$, aligns with the boundary edge direction to discourage simplification along sharp boundary curves. The total boundary quadric is constructed by summing the base quadrics of both planes, scaled by the estimated curvature $\kappa$ as follows:

$$\mathbf{p}_1 = [\mathbf{n}_1, -\mathbf{n}_1 \cdot \mathbf{v}_1], \quad \mathbf{n}_1 = (\mathbf{v}_1 - \mathbf{v}_2) \times (\mathbf{v}_3 - \mathbf{v}_1) \quad (7)$$

$$\mathbf{p}_2 = [\mathbf{d}, -\mathbf{d} \cdot \mathbf{v}_1], \quad \mathbf{d} = \mathbf{v}_1 - \mathbf{v}_2 \quad (8)$$

$$Q_{boundary} = w_{boundary} \cdot \kappa \cdot (\mathbf{p}_1\mathbf{p}_1^T + \mathbf{p}_2\mathbf{p}_2^T) \quad (9)$$

This discourages collapses in regions of high boundary curvature, helping preserve sharp features and edge integrity. The dual-plane constraint forms a stronger "splint" than prior approaches such as QEM4VR [1], which rely on a single, arbitrarily oriented penalty plane and thus offer weaker protection against feature flattening.

**Normal Preservation** ($Q_{normal}$): To retain high frequency surface details from the original mesh we penalize deviations from the original mesh vertices normal. Precisely, for each vertex $\mathbf{v}_k$ with unit normal $\mathbf{n}_k = [n_x, n_y, n_z]^T$, we define a tangent plane $\mathbf{p}$ that passes through $\mathbf{v}_k$ as $\mathbf{p} = [n_x, n_y, n_z, -n_k \cdot \mathbf{v}_k]^T$. We then compute the total penalty cost ($Q_{normal}$) for preserving this tangent plane. This penalty penalizes any new vertex position $\mathbf{v}'$ that deviates from this original tangent plane, restricting movement along the normal and thus reducing faceting artifacts.

$$Q_{normal} = w_{normal} \cdot \mathbf{p}\mathbf{p}^T \quad (10)$$

### 3.2.2. Area Preservation Cost

3D models with significant open boundaries are common in VR and gaming assets. Preserving the shape and extent of the model's rim is essential to maintain silhouette and topological fidelity. A common failure mode in mesh simplification is the excessive collapse of boundary edges, which can visually shrink or distort the boundary regions. To address this, we adapt the area-based boundary preservation quadric proposed by Lindstrom and Turk [15], which penalizes edge collapses that significantly reduce the local boundary area.

Unlike the standard quadrics that measure deviation from surface planes, the area quadric encodes the change in surface area introduced by collapsing a boundary edge. This area preservation cost ($cost_{area}$), is computed independently and added to the total cost (refer equation 2). The computation proceeds as follows: given a candidate edge $(\mathbf{v}_i, \mathbf{v}_j)$ and estimated collapse point $\mathbf{v}'$, we construct total area preservation quadric $Q^A$ from the one-ring neighborhood of $\mathbf{v}_i$ and $\mathbf{v}_j$, but only considering their incident boundary edges (i.e., edges shared by a single face). For each boundary edge $(\mathbf{v}_r, \mathbf{v}_s)$, where $r \in \{i, j\}$ and $s$ is a boundary neighbor, we define an *edge vector*: $\mathbf{e}_{rs} = \mathbf{v}_s - \mathbf{v}_r$, *triangle vector*: $\mathbf{t}_{rs} = \mathbf{v}_r \times \mathbf{v}_s$, and *cross-product matrix*: $E = [\mathbf{e}_{ab}]_\times$, where

$$[\mathbf{x}]_\times = \begin{bmatrix} 0 & -x_z & x_y \\ x_z & 0 & -x_x \\ -x_y & x_x & 0 \end{bmatrix}$$

The total area quadric $Q^A$ for an edge can be computed

as below, where $Q_{rs}$ is the per-edge area quadric.

$$Q^A = \sum_{(r,s)} Q_{rs}, \quad Q_{rs} = \frac{1}{2} \begin{bmatrix} E^T E & -E\mathbf{t}_{rs} \\ -(E\mathbf{t}_{rs})^T & \mathbf{t}_{rs}^T \mathbf{t}_{rs} \end{bmatrix} \quad (11)$$

Finally, we compute the boundary area preservation cost for the proposed vertex $\mathbf{v}' = [x, y, z, 1]^T$ as $\text{cost}_{area}(\mathbf{v}') = \mathbf{v}'^T Q^A \mathbf{v}'$. This additional term discourages collapses that would significantly reduce or distort the rim area of the mesh. The influence of this cost is controlled via the user-defined weight $w_{area}$ in equation 2.

### 3.2.3. Texture Transfer via Successive Mapping

Once we obtain the geometrically simplified mesh $M_{simplified}$, our objective is to accurately transfer appearance from the original mesh $M_{original}$. This step is particularly important in modern 3D pipelines, where high-resolution textures from reconstruction or generative models must be preserved after aggressive geometric simplification. To achieve this, we adopt a simplified version of the robust successive mapping method [17], which enables stable and high-fidelity appearance transfer.

$$T : M_{simplified} \rightarrow M_{original} \quad (12)$$

**Our Successive Mapping Process:** During simplification, we record the full sequence of edge collapses, forming a progressive history. The `successive_mapping` function leverages this history by iteratively reversing each collapse, starting from the final simplified mesh $M_{simplified}$ and working backward to the original mesh $M_{original}$.

At each $i^{th}$ step of successive mapping, a vertex in $M_{simplified}^{i+1}$ is *un-collapsed* into its two parent vertices in $M_{simplified}^i$. Rather than performing a computationally expensive geometric projection of sample points onto the restored surface, we apply a simple and effective heuristic: each point associated with the collapsed vertex is reassigned to the closer of the two parent vertices. This step-wise, local reversal ensures that the mapping remains spatially consistent with the original simplification path. In contrast, direct one-shot projection methods can fail on thin structures or complex geometry [12, 17], e.g., incorrectly mapping a point to the opposite side of a thin wall. Our successive strategy avoids such errors by preserving the collapse lineage, producing mappings that are both stable and locality-preserving.

**Final Texture Baking:** Once the correspondence map $T$ is established, we generate a new texture for the simplified mesh by sampling its surface and querying appearance from the original mesh. For each point $p \in M_{simplified}$, we evaluate $T(p) \in M_{original}$ and sample the corresponding color from the original texture. The resulting appearance data is then baked into a new texture atlas using standard parameterization techniques, e.g., per-triangle flattening [32], or optionally encoded as per-vertex color attributes.

By eliminating reliance on the original UV layout, our approach avoids texture bleeding artifacts, resolves issues at UV seams [1], and offers higher texture fidelity on aggressively simplified meshes.

### 3.3. Topological Integrity and Robustness

Our method is designed to handle *meshes in the wild*, a term encompassing meshes that may exhibit a variety of complex conditions. Figure 3 illustrates several of these key geometric features, including non-manifold vertices, open boundaries, and disconnected components, which our algorithm is built to address. We incorporate several key strategies to ensure topological robustness during simplification:

**Normal Flip Prevention:** Before committing to any edge collapse, we validate the operation by checking for surface orientation reversals. For each face adjacent to the collapsing edge, we compute the triangle normals before and after the collapse. If the dot product between any corresponding pair of normals is negative, indicating a potential flip or inversion, the collapse is rejected. This prevents artifacts such as inverted normals and ensures the resulting surface maintains correct orientation.

**Support for Non-Manifold Geometry:** Our mesh data structures and collapse procedure are designed to handle non-manifold conditions. We maintain generalized vertex-to-vertex and vertex-to-face adjacency without assuming manifoldness. This enables our algorithm to operate reliably on real-world assets, including those with T-junctions, self-intersections, and duplicated edges.

**Virtual Edge Insertion for Component Merging:** To simplify meshes with multiple disconnected components, we optionally introduce *virtual edges* between spatially proximate components. These edges are not part of the original mesh topology but serve as collapse candidates. Proximity is determined by a distance threshold relative to the model's bounding box diagonal, and several safeguards prevent the creation of degenerate zero-length edges, as detailed in our supplementary material. This encourages merging of disconnected fragments into a single, cohesive low-poly mesh, improving both compactness and downstream usability in applications such as simulation, digital twins, and large-scale 3D processing pipelines.

## 4. Experiments

To validate the effectiveness, robustness, and practical utility of FA-QEM in modern 3D pipelines, we conduct a comprehensive set of experiments. Our evaluation is designed to assess performance on both real-world and generated 3D assets, reflecting the challenges posed by contemporary reconstruction and generative methods. Specifically, we use two complementary datasets: (1) For geometric accuracy, robustness, and scalability, we use the challenging Thingi10K dataset [34], which contains diverse, noisy, and

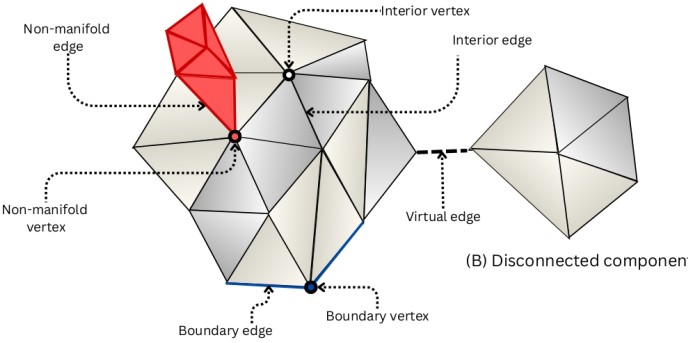

**(B) Disconnected component**

**(A) Main Mesh Component**

Figure 3. **Key mesh terminology.** A *boundary edge* is incident to one face, a *non-manifold edge* is shared by more than two faces, and a *virtual edge* is an artificial collapse candidate used to merge nearby disconnected components.

Table 1. **Hyperparameters used in all experiments.** These fixed weights control geometric fidelity, boundary preservation, area regularization, and normal alignment in the FA-QEM composite quadric, and are kept constant across datasets.

| Parameter | Value | Purpose |
|---|---|---|
| $w_{area}$ | 100.0 | Weight for boundary area preservation cost. |
| $w_{boundary}$ | 500.0 | Scales the curvature-based boundary penalty. |
| $w_{uv}$ | 5000.0 | Multiplicative penalty for vertices on UV seams. |
| $w_{normal}$ | 0.01 | Weight for the normal preservation quadric. |
| $w_{plane\_area}$ | 1.0 | Controls the strength of inverse-area weighting. |

non-manifold meshes representative of real-world and user-generated content; (2) for appearance preservation, we use the Real-World Textured Things dataset [21], which contains high-resolution textured scans suitable for evaluating texture fidelity. This setup allows us to rigorously evaluate FA-QEM as a component for converting dense, unstructured meshes into efficient, high-quality 3D assets.

We compare FA-QEM against representative classical and state-of-the-art methods on the above datasets and also on AI generated meshes [29, 33]: QEM [5], QEM4VR [1], ICE [16], RobustLPM [2], CWF [30], and Liu et al. (STMW) [17]. All experiments are conducted on a CPU to demonstrate the method's efficiency without reliance on GPU acceleration.

To ensure a fair and reproducible comparison, all hyperparameters (refer Table 1) for FA-QEM were kept fixed across all datasets and experiments, demonstrating the general applicability of our method. A detailed breakdown of these parameters is provided in the supplementary material.

### 4.1. Geometric Evaluation and Performance

A primary design goal of FA-QEM is to robustly simplify complex, real-world meshes while maintaining high geometric fidelity and efficiency. We evaluate these properties on the Thingi10K dataset [34], which serves as a challenging benchmark due to its prevalence of non-manifold ge-

ometry, noise, and irregular topology. These characteristics closely resemble meshes produced by modern reconstruction and generative pipelines. Our method achieves a 100% success rate, successfully processing all 10,000 meshes down to 1% of their original face count without failure, demonstrating strong robustness in real-world scenarios.

The geometric error metrics in Table 2 further show that this robustness does not come at the cost of fidelity. The qualitative examples in Figure 4 are drawn directly from this challenging dataset and serve as visual substantiation of our method's ability to handle complex and irregular inputs.

Beyond robustness, we evaluate geometric fidelity and speed. We decimate each mesh to 10% and 1% of its original resolution and report the symmetric Hausdorff distance, mean squared Chamfer distance, and average runtime. As shown in Table 2, FA-QEM consistently achieves the lowest geometric error, indicating superior preservation of shape and fine details, and enabling the generation of compact, high-fidelity meshes suitable for downstream applications.

We further compare with the recent Robust Low-Poly Meshing (LPM) method [2]. Since the implementation released by the authors only runs on a specific subset of 100 models of Thingi10K dataset, a direct comparison within our main benchmark was not possible. We therefore compare FA-QEM on this same subset. On this specific subset, our FA-QEM outperforms Robust LPM, achieving Hausdorff/Chamfer errors of 0.0050/0.00043 compared to their 0.0280/0.00102. We attribute the performance gain due to the difference in the methodologies i.e., RobustLPM is a remeshing algorithm that generates a new mesh topology, while our decimation-based FA-QEM preserves the original topology as much as possible.

Critically, this high fidelity is achieved with a significant performance advantage. Table 2 shows that FA-QEM is substantially faster than other robust and feature-aware techniques like Liu et al. (STMW) [17], CWF [30] and QEM4VR [1]. This efficiency is attributed to a synergy between our streamlined quadric formulation and an optimized implementation. Our method is designed to translate complex feature goals into efficient linear algebraic operations, and our use of JIT-compilation for numerical hotspots ensures this theoretical efficiency is realized in practice. A detailed runtime profile is shown in Table 6. A further detailed discussion is provided in the supplementary material. Qualitative comparisons in Figure 4 further illustrate that FA-QEM preserves sharp features and avoids the geometric and textured artifacts seen in other methods. This efficiency makes FA-QEM particularly well-suited for integration into large-scale 3D processing pipelines where both speed and fidelity are critical.

Our method's robustness extends to a variety of challenging cases, including large-scale meshes, noisy scans, and

Table 2. **Quantitative results on Thingi10K [34].** We report average Hausdorff and Chamfer distances at 10% and 1% resolutions. FA-QEM achieves the lowest geometric error while also providing significantly faster runtime than prior quadric-based and feature-preserving methods.

| Method | 10% Resolution | | | 1% Resolution | |
| | $Hausdorff(\times 10^{-1})\downarrow$ | $Chamfer(\times 10^{-1})\downarrow$ | $Time(s)\downarrow$ | $Hausdorff(\times 10^{-1})\downarrow$ | $Chamfer(\times 10^{-1})\downarrow$ |
|---|---|---|---|---|---|
| QEM [5] | 0.13 | 0.4600 | **4.25** | 0.57 | 1.3700 |
| QEM4VR [1] | 0.78 | 0.0136 | 29.01 | 2.75 | 0.1386 |
| ICE [16] | 2.71 | 0.2815 | 16.55 | 4.25 | 0.8254 |
| CWF [30] | 0.66 | 0.0189 | 21.77 | - | - |
| Liu et. al (STMW) [17] | 0.10 | 0.2900 | 37.70 | 0.44 | 1.1100 |
| FA-QEM | **0.08** | **0.0072** | 10.60 | **0.25** | **0.0173** |

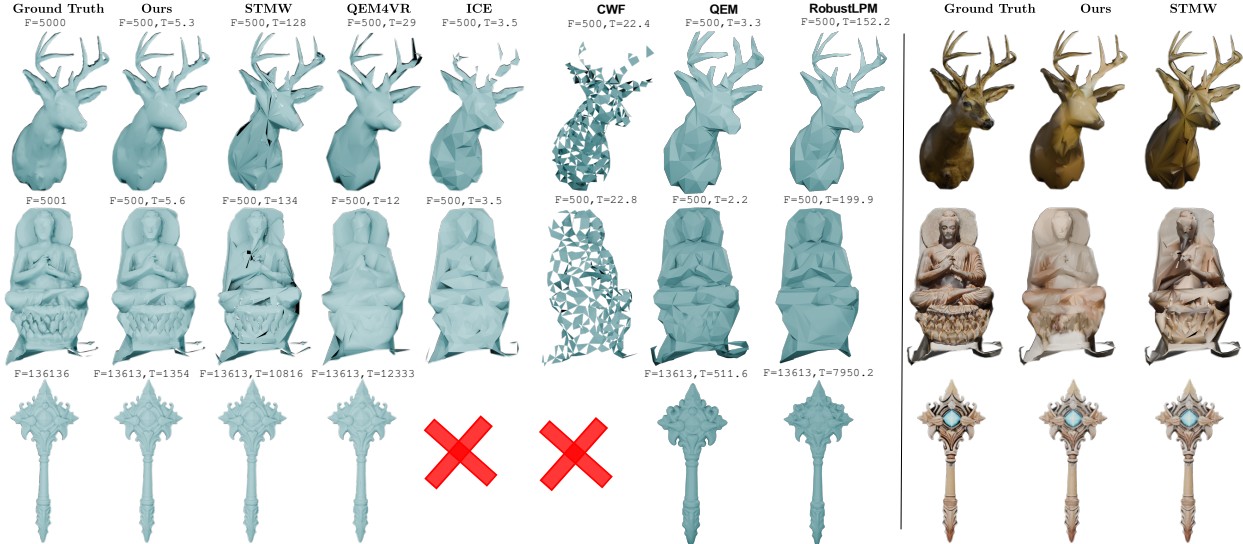

Figure 4. **Qualitative comparison with state-of-the-art methods.** We compare geometric and textured meshes simplified to a fixed face count $F$. $T$ denotes runtime (s), and a red **X** indicates failure to reach the target resolution. Top two rows are from the Real-World Textured Things dataset [21], and the bottom row shows an AI-generated mesh [33]. FA-QEM preserves fine geometry and texture under aggressive simplification while achieving lower runtimes. See supplementary for additional results.

Table 3. **Runtime breakdown of FA-QEM on the mesh in Fig. 1.** Most computation occurs in the iterative edge collapse loop, dominated by vertex placement and neighborhood updates. Percentages are normalized to total runtime.

| Component | % of Total Runtime |
|---|---|
| Initial Quadric Construction (Sec 3.2.1) | 25% |
| Priority Queue Population | 15% |
| Iterative Collapse Loop | 55% |
|   *- Edge Pop from Queue* | *5%* |
|   *- Optimal Position Solve* | *20%* |
|   *- Area Cost Calculation* | *10%* |
|   *- Neighbor Updates* | *20%* |
| Successive Mapping & Texture Bake | 5% |

models with disconnected components. We provide a more detailed breakdown of our method's performance on these challenging cases, including its handling of non-manifold geometry and noisy scans, in Section 10 of the supplementary material.

## 4.2. Texture Preservation Evaluation

To assess appearance preservation, which is critical for modern reconstruction and generative pipelines, we evaluate texture fidelity using the Real-World Textured Things dataset [21]. We focus on a subset of approximately 400 meshes with a single texture image and simplify each mesh to 1% of its original resolution. We measure texture quality using the symmetric Chamfer distance on textures [31], which captures perceptual fidelity independent of viewpoint.

As our method generates new texture coordinates via successive mapping, we compare only against methods that follow a similar attribute transfer philosophy. FA-QEM achieves a texture error of 0.099. This result is highly competitive with the state-of-the-art method by Liu et al. (STMW) [17], which scores 0.078. This slight gap is expected, as Liu et al. (STMW) [17] directly optimizes for texture consistency, whereas our method prioritizes geometric fidelity, leading to a better overall speed-quality trade-

Table 4. **Ablation study of FA-QEM components.** Each row incrementally adds one term to the composite metric. We report geometric (Hausdorff, Chamfer) and texture Chamfer errors at 10% resolution. Each component improves performance, with the full model achieving the best results.

| Configuration | Geometric Hausdorff $(\times 10^{-1})(\downarrow)$ | Geometric Chamfer $(\times 10^{-1})(\downarrow)$ | Texture Chamfer $(\times 10^{-1})(\downarrow)$ |
|---|---|---|---|
| (1) FA-QEM (w/ all w's=0) | 0.1970 | 0.0267 | 0.1141 |
| (2) + $w_{area}$ | 0.1404 | 0.0173 | 0.1115 |
| (3) + $w_{plane\_area}$ | 0.1230 | 0.0135 | 0.1109 |
| (4) + $w_{boundary}$ | 0.0910 | 0.0084 | 0.1035 |
| (5) Full FA-QEM (+ $w_{normal}$) | **0.0800** | **0.0072** | **0.0990** |

off.

While their error is slightly lower, our method is over 3.5x faster (10.60s vs 37.70s, as shown in Table 2), offering a significantly more practical trade-off between speed and fidelity for real-world 3D pipelines. By decoupling geometry from the original UV layout and using a robust successive mapping strategy, our method effectively mitigates the stretching and bleeding artifacts that constrain traditional UV-preserving approaches.

## 4.3. Ablation Study

To understand the individual contribution of each term in our composite cost function, we conduct a cumulative ablation study. Starting from FA-QEM with all weights equal to zero, we incrementally enable each proposed component and measure the resulting geometric and texture error. The results, summarized in Table 4, demonstrate the clear benefit of each component. The introduction of boundary area ($w_{area}$), area weighting ($w_{plane\_area}$), and our dual-plane boundary preservation ($w_{boundary}$) progressively reduces geometric error. The addition of the normal preservation term ($w_{normal}$) provides the final significant improvement in both Hausdorff and Chamfer distance. The full FA-QEM model, integrating all components, achieves the best overall performance, confirming that our joint optimization leads to superior simplification in both geometry and appearance.

Our findings confirm each component's value. Boundary area preservation ($w_{area}$) reduces error on non-watertight models. Inverse area weighting ($w_{plane\_area}$) and normal preservation ($w_{normal}$) improve fidelity in high-detail regions. Finally, our dual-plane boundary preservation ($w_{boundary}$) preserves sharp creases and corners.

To further validate our composite metric and address the rigor of our design choices, we conducted a sensitivity analysis on our key hyperparameters. Our analysis shows the impact of varying $w_{boundary}$, the weight for our novel curvature preservation quadric (Sec. 3.2.1). This parameter is central to our "Feature-Aware" contribution.

The graph plots geometric error (Hausdorff and Chamfer) against the parameter value. The error is high when

$w_{boundary} = 0$, confirming the findings of our ablation study (Table 4). The error metrics drop to a distinct minimum at or near our chosen value of 500. Importantly, the "U-shaped" curve is shallow around this optimum, indicating that our method is robust and not overly sensitive to this parameter.

We conducted an identical analysis for our other key weights, $w_{area}, w_{normal}$, and $w_{plane\_area}$. All parameters exhibited similar stable, U-shaped curves, confirming their contribution. For completeness, the full sensitivity analysis, including all graphs, is provided in the supplementary material.

### 4.4. Limitations and Failure Cases

While FA-QEM demonstrates strong robustness on the challenging Thingi10K dataset, our method is not without limitations.

**Complex Topologies:** Our method can handle disconnected components via virtual edge insertion. However, some scenarios remain challenging, such as meshes with extremely thin, interleaved, or self-intersecting surfaces. For example, a model of two chain-link fences passing through each other may cause our virtual edge heuristic to create ambiguous connections.

**Texture Mapping:** Our texture transfer relies on the successive mapping heuristic from [17]. While effective, this mapping could potentially be improved with higher-precision projections for highly contorted regions. In such pathological cases, the successive mapping could incorrectly associate points from one surface with another across a thin gap. We identify these specific, complex cases as an avenue for future work.

Overall, these results demonstrate that FA-QEM provides a robust, efficient, and high-fidelity solution for simplifying complex meshes, making it a practical and effective component for modern 3D reconstruction and generative pipelines.

## 5. Conclusion

We introduced FA-QEM, a fast, robust, and feature-aware mesh simplification pipeline that addresses the trade-off between performance and visual fidelity. Our key contribution is a composite Quadric Error Metric that jointly models geometric deviation, boundary curvature, and normal consistency, enabling high-quality simplification of noisy and non-manifold meshes. We further show that improved geometric simplification enhances texture transfer via successive mapping. Experiments on real-world and AI-generated datasets demonstrate state-of-the-art geometric and textural fidelity with significantly lower computational cost. Overall, FA-QEM provides a scalable solution for converting dense meshes into compact, high-quality assets, enabling efficient use in modern 3D reconstruction, generative pipelines, and downstream applications.

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

# Fast and Robust Mesh Simplification for Generated and Real-World 3D Assets

## Supplementary Material

## 6. Implementation Details

This supplementary document provides additional implementation and experimental details for FA-QEM, a feature-aware mesh simplification pipeline designed for modern 3D reconstruction and generative workflows. Our implementation focuses on scalability, robustness, and efficiency, enabling the conversion of dense, unstructured meshes into compact, high-quality geometric assets suitable for downstream applications.

Our implementation is written in Python and leverages several open-source libraries, including Open3D for mesh data structures, NumPy and SciPy for numerical operations, and Numba for performance-critical computations.

Please refer to the supplementary video for a $360°$ visualization of the simplified meshes.

### 6.1. Core Algorithm and Data Structures

Our method is built upon an iterative edge collapse framework. The core data structures are:

**Adjacency Information:** We pre-compute and maintain vertex-to-vertex ($v\_to\_v$) and vertex-to-face ($v\_to\_t$) adjacency maps using Python dictionaries. This allows for efficient queries of the one-ring neighborhood of a vertex, which is essential for all quadric and cost calculations.

**Priority Queue:** We use a min-priority queue, implemented with Python's heapq module, to store all valid edges in the mesh. Each entry in the queue contains the edge's collapse cost, the optimal target position for the new vertex ($v'$), and a unique identifier to handle potential updates. The simplification process proceeds by iteratively extracting the edge with the lowest cost from the queue, performing the collapse, and updating the costs of all neighboring edges affected by the topological change.

### 6.2. Virtual Edge Insertion and Degeneracy Handling

This subsection provides a detailed breakdown of the virtual edge insertion process and the safeguards our implementation uses to handle potential geometric degeneracies, as mentioned in Sec. 3.3 of the main paper.

**Virtual Edge Insertion Criteria:** The virtual edge insertion is an optional pre-processing step designed to merge disconnected but spatially coherent components. The process is as follows:

1. **Component Identification:** We first identify all topologically disconnected components in the mesh using a standard breadth-first search on the mesh graph.

2. **Proximity Search:** For each component, we construct a KD-Tree of its triangle centroids. We then perform a ball query between the KD-Trees of different components to find candidate pairs of triangles whose centroids are within a user-defined proximity threshold $\tau$. To ensure scale invariance, we define this threshold as $\tau = 0.01 \cdot L_{diag}$, where $L_{diag}$ is the length of the mesh's bounding box diagonal.

3. **Candidate Edge Creation:** For each pair of proximate triangles, we find the two closest vertices between them. This vertex pair becomes a "virtual edge" candidate and is added to our initial set of edges for the priority queue. Its cost is computed in the same manner as a standard topological edge.

**Safeguards Against Degeneracies:** Our implementation has several safeguards against the risk of creating zero-length virtual edges from coincident or overlapping faces:

**Pre-emptive Vertex Merging:** Before simplification, our mesh loading process includes a standard pre-processing step that merges all vertices that are closer than a small absolute tolerance of 1e-6. This automatically welds many coincident faces from the input model.

**Edge Collapse Validation:** During the cost calculation, any edge (virtual or real) that has a length below a small relative tolerance is assigned an effectively infinite cost, preventing it from ever being selected from the priority queue. This threshold is set to 1e-8 times the length of the mesh's bounding box diagonal, making the check robust to models of different scales.

**Numerical Stability:** Our linear system solver for finding the optimal vertex position is robust to degenerate quadrics. In cases where the matrix is singular (which can be caused by co-planar geometry), our method falls back to selecting one of the endpoints or the midpoint, a stable operation even for zero-length edges.

These safeguards work in concert to prevent degeneracies from affecting the stability of the simplification process.

## 7. Hyperparameter Justification

The performance of FA-QEM is governed by a small set of weighting parameters. Importantly, all hyperparameters used in the main paper are fixed across datasets and models, demonstrating that FA-QEM operates as a general-purpose method without requiring per-instance tuning.

The values were determined empirically by testing on a separate validation set of 100 diverse models from both the Thingi10K and the Real-World Textured Things dataset [21,

Table 5. Hyperparameter values used for all experiments.

| Parameter | Value | Purpose |
|---|---|---|
| $w_{area}$ | 100.0 | Weight for boundary area preservation cost. |
| $w_{boundary}$ | 500.0 | Scales the curvature-based boundary penalty. |
| $w_{uv}$ | 5000.0 | Multiplicative penalty for vertices on UV seams. |
| $w_{normal}$ | 0.01 | Weight for the normal preservation quadric. |
| $w_{plane\_area}$ | 1.0 | Controls the strength of inverse-area weighting. |

34], which were not included in our final evaluation set. The goal of this tuning process was to find a robust balance between geometric, feature, and attribute preservation that would generalize well. For example, the high value for $w_{uv}$ reflects a strong prior that collapsing across texture seams is almost always undesirable, while the smaller value for $w_{normal}$ provides a gentle constraint to maintain smooth shading without overly restricting geometric optimization.

The fixed values used for all experiments are detailed in Table 5.

## 7.1. Sensitivity Analysis

To validate our chosen hyperparameters, we performed a comprehensive sensitivity analysis. We evaluated the impact of each key weight in our composite cost function: $w_{boundary}$, $w_{area}$, $w_{normal}$, and $w_{plane\_area}$.

**Experimental Setup** We conducted these experiments on a diverse validation subset of 100 meshes from both the Thingi10K and the Real-World Textured Things dataset [21, 34], distinct from the test set used for our main results. For each experiment, we varied one parameter across a wide range of values while keeping all other parameters fixed to the default values reported in Table 1 of the main paper. We measured the average geometric error (Hausdorff and Chamfer distance) for the simplified meshes at 10% resolution.

**Analysis of Results** The results are plotted in Figure 5 through Figure 8.

**Boundary Weight ($w_{boundary}$):** As shown in Figure 5, setting this weight to 0 results in high error, confirming the necessity of the term. The error drops significantly as the weight increases, reaching a stable minimum around our chosen value of 500. The curve is notably shallow around the optimum, indicating that the method is robust to deviations in this parameter (e.g., values between 250 and 750 yield similar performance).

**Area Preservation ($w_{area}$):** Figure 6 demonstrates that the area term is crucial for preventing volume loss and silhouette degradation. The error decreases rapidly as $w_{area}$ is introduced and remains stable for values $> 50$, confirming our choice of 100 is safe and effective.

**Normal Preservation ($w_{normal}$):** Figure 7 shows that even a small weight (0.01) significantly improves geometric fidelity by preventing faceting artifacts. Excessive values can over-constrain the simplification, but the method

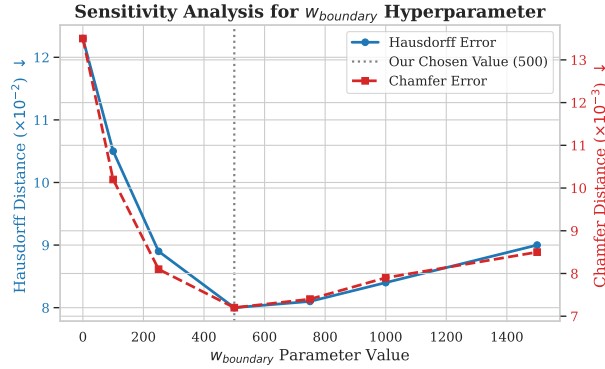

Figure 5. **Sensitivity Analysis:** $w_{boundary}$. The error drops to a clear minimum around 500. The stable "U-shape" confirms the parameter is well-tuned and robust.

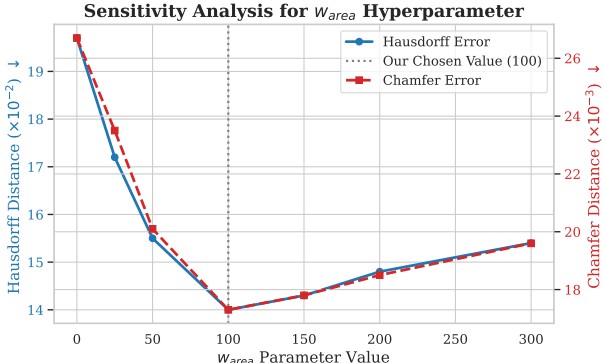

Figure 6. **Sensitivity Analysis:** $w_{area}$. Introduction of the area term significantly reduces error compared to the baseline (0), with stable performance observed for values above 50.

remains stable within a reasonable range.

**Inverse Area Weighting ($w_{plane\_area}$):** Figure 8 validates the benefit of penalizing collapses in high-curvature regions more heavily than flat regions.

Overall, these analyses demonstrate that FA-QEM is not sensitive to precise parameter tuning. The performance remains stable across a wide range of values, and the selected configuration represents a robust operating point that generalizes well to diverse, real-world meshes.

## 8. Performance and Scalability

FA-QEM is designed for efficient processing of large-scale 3D assets arising from reconstruction and generative pipelines. Its performance advantage over prior methods stems from both algorithmic design and implementation-level optimizations.

**Algorithmic Contributions to Performance:** Our primary algorithmic performance gain comes from the design of our

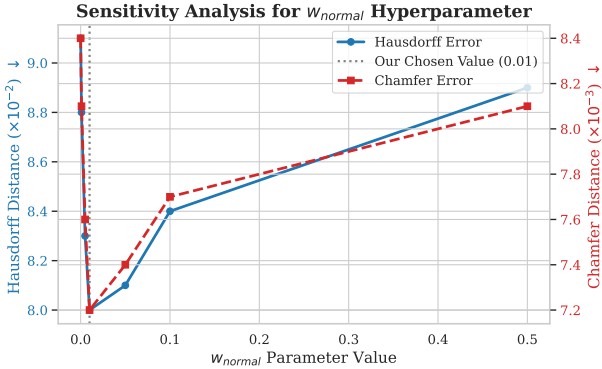

Figure 7. **Sensitivity Analysis:** $w_{normal}$. A small weight of 0.01 provides the optimal balance, reducing faceting without over-constraining the geometry.

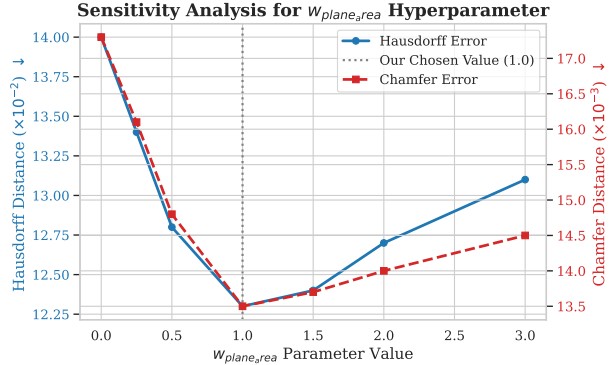

Figure 8. **Sensitivity Analysis:** $w_{plane\_area}$. The inverse-area weighting effectively guides simplification to flat regions, minimizing error at our chosen value of 1.0.

decoupled, 2-stage cost metric.

1. **Efficient Optimal Position Search:** By creating a single, composite geometric-feature quadric $(Q_{gf})$ upfront, we only need to solve one small $(3 \times 3)$ linear system per edge to find the optimal vertex position. This is computationally much cheaper than methods that might involve iterative searches or more complex, higher-dimensional optimization spaces to balance multiple constraints.

2. **Streamlined Cost Calculation:** The subsequent calculation of the boundary area cost $(cost_{area})$ is also a direct evaluation on a pre-computed quadric $(Q^A)$. This avoids costly on-the-fly geometric queries (e.g., recalculating local areas for every potential collapse) that can be a bottleneck in other methods.

Our formulation expresses complex feature-preserving objectives as efficient linear algebraic operations, enabling fast and scalable computation.

**Implementation-Level Optimizations:** We complement our algorithmic design with several key implementation

Table 6. Runtime profile for FA-QEM on a representative mesh.

| Component | % of Total Runtime |
|---|---|
| Initial Quadric Construction (Sec 3.2.1) | 25% |
| Priority Queue Population | 15% |
| Iterative Collapse Loop | 55% |
| *- Edge Pop from Queue* | *(5%)* |
| *- Optimal Position Solve* | *(20%)* |
| *- Area Cost Calculation* | *(10%)* |
| *- Neighbor Updates* | *(20%)* |
| Successive Mapping & Texture Bake | 5% |

choices:

**Pre-computed Adjacency:** We pre-cache all vertex-to-face and vertex-to-vertex adjacencies, allowing for $O(1)$ lookups of one-ring neighborhoods.

**Optimized Numerical Kernels:** Critical functions, such as curvature estimation and the initial quadric summation, are JIT-compiled using Numba for highly efficient execution.

**Efficient Data Structures:** We use Python's $heapq$ for the priority queue and dictionaries for adjacency maps, which are proven choices for performance.

**Runtime Profile Breakdown:** To provide concrete data, we profiled the execution of FA-QEM on a representative 100k-face model from the Real-World Textured Things dataset [21]. The breakdown of the total runtime is shown in Table 6.

This profile shows a balanced distribution of work. The significant time spent in the initial quadric construction (25%) reflects our algorithmic choice to pre-process features, while the speed of the iterative loop is a result of both our efficient cost formulation (fast position solve and area cost) and our implementation-level optimizations. This synergy is the key to FA-QEM's overall performance.

**Large-Scale Meshes:** To validate performance on large-scale assets, we profiled FA-QEM on meshes of varying complexity ranging from 5k to 240k faces. All experiments were conducted on a standard consumer CPU (Intel Core i5-1135G7 @ 2.40GHz). As illustrated in Figure 9, our runtime exhibits polynomial scaling with respect to the input face count. While the complexity grows for very large meshes due to the Python-based implementation overhead, FA-QEM remains substantially more efficient than competing robust methods. For the high-resolution 'Lamp' model (240k faces), our method completed simplification to 24k faces in 1,710 seconds, whereas the state-of-the-art robust method Liu et al. (STMW) [17] required 25,201 seconds. This substantial speedup demonstrates that FA-QEM provides a practical and scalable solution for processing large-scale 3D assets in real-world pipelines.

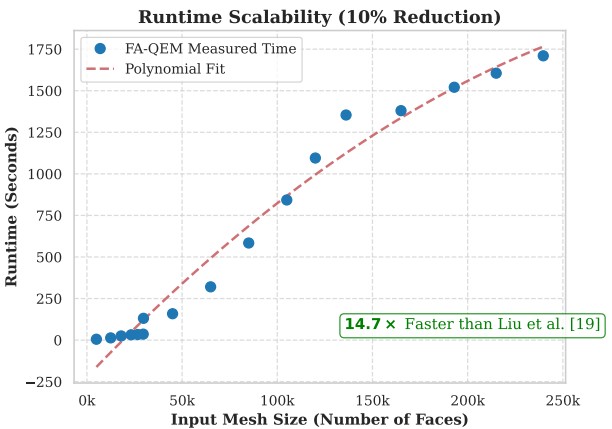

Figure 9. **Runtime Scalability Analysis.** We plot the execution time of FA-QEM against input mesh size on a standard Intel Core i5 CPU. The method exhibits consistent polynomial scaling ($O(N^2)$), verifying its stability across varying mesh complexities. Notably, for the high-resolution 'Lamp' model (240k faces), FA-QEM completes in 28 minutes, achieving a $14.7\times$ speedup over the state-of-the-art robust method Liu et al. (STMW) [17].

## 9. Detailed Texture Mapping Validation

We further evaluate the effectiveness of our texture transfer strategy, particularly in the context of modern 3D pipelines where high-quality appearance must be preserved after aggressive geometric simplification.

### 9.1. Quantitative Trade-off: Efficiency vs. Fidelity

In the main paper, we noted that FA-QEM achieves texture quality highly competitive with Liu et al. (STMW) [17] while being significantly faster. To substantiate this, we provide a direct comparison of Runtime vs. Texture Error (Chamfer) for 9 representative models from our test set in Figure 10.

As illustrated, FA-QEM (blue circles) consistently occupies the high-efficiency region of the plot. While Liu et al. (STMW) [17] (red crosses) achieves marginally lower error on some models, it requires orders of magnitude more time (note the log scale). For example, on the 'Lamp' model, Liu et al. (STMW) [17] requires over 7 hours to achieve a Texture Chamfer error of 0.108, while FA-QEM achieves a comparable 0.146 in just 28 minutes. This confirms that FA-QEM offers a far superior trade-off for production environments where time is a constraint.

### 9.2. Why Successive Mapping Beats Projection

Our chosen method (successive mapping) is superior to standard ray-casting or nearest-neighbor projection, particularly in failure cases.

**The Projection Problem:** Standard texture transfer often involves casting rays from the simplified surface to the

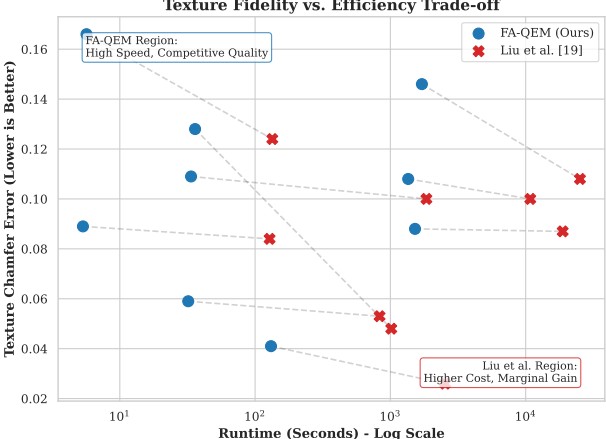

Figure 10. **Efficiency vs. Fidelity Trade-off.** We compare FA-QEM against the state-of-the-art texture preserving method Liu et al. (STWM) [17]. Connecting dashed lines link the same underlying mesh. FA-QEM offers orders-of-magnitude faster processing (left side) for comparable texture quality (vertical axis).

original to sample colors. This can fail in challenging scenarios in "thin-wall" scenarios (e.g., a sword blade or a folded cloth). If the simplified mesh deviates even slightly, the ray may miss the intended surface and hit the back-face or a surface behind it, sampling the wrong color and causing severe artifacts.

**The Successive Advantage:** Our approach avoids geometric spatial search entirely. Instead, we track the *history* of edge collapses. When an edge $(\mathbf{u}, \mathbf{v})$ collapses to $\mathbf{v}'$, the texture coordinates for $\mathbf{v}'$ are derived directly from $\mathbf{u}$ and $\mathbf{v}$ based on local proximity in the graph. This preserves the topological neighborhood. Even if a "thin wall" collapses, the texture mapping remains logically attached to the correct side of the surface, preventing the "bleed-through" artifacts common in projection methods.

## 10. Robustness and Limitations

This section provides a more detailed analysis of FA-QEM's performance on the challenging cases, substantiating the robustness claims made in the main paper.

**Large Texture-Varying Inputs:** Our method is particularly well-suited for models with complex texture layouts due to our philosophy of decoupling geometry from the UV parameterization. Methods [1, 4] that strictly preserve UVs are unable to simplify across the boundaries of different texture islands or charts. In contrast, our successive mapping technique is agnostic to the texture content itself. It only requires a valid geometric correspondence, allowing it to correctly pull color from any part of the original texture atlas. This allows FA-QEM to gracefully handle models with many texture islands, simplifying the geometry optimally

without being constrained by the texture layout.

**Noisy Scans and Highly Irregular Meshes:** As stated in the main paper, our method achieved a 100% processing success rate on the 10,000-model Thingi10K dataset [34]. This dataset is known to contain numerous models derived from noisy scans or authored with topological errors. Our robustness stems from two key design choices. First, the QEM [5] framework itself is inherently noise-tolerant, as the summation of plane quadrics has a natural averaging effect that smooths high-frequency noise. Second, our algorithm is designed from the ground up to handle irregular meshes by not assuming manifold topology. Our generalized adjacency data structures and the crucial normal-flip veto (Sec. 3.3) ensure that collapses in challenging local geometric configurations (e.g., at T-junctions) do not create inconsistent geometry.

**Disconnected Components and Failure-Case Analysis:** Our method handles disconnected components via the optional virtual edge insertion phase (Sec. 3.3). However, we acknowledge that our method is not without limitations. A challenging scenario, which represents a potential failure case, is the simplification of extremely thin, interleaved, or self-intersecting surfaces. For example, consider a model of two chain-link fences passing through each other. Our virtual edge heuristic may create ambiguous connections, and the successive mapping could incorrectly associate points from one surface with the other across the thin gap. In such pathological cases, pre-processing via semantic segmentation would be a more appropriate strategy. We identify this as an avenue for future work.

These observations highlight both the robustness of FA-QEM in practical scenarios and the limitations that arise in extreme geometric configurations, which we identify as directions for future work.

## 11. Additional Qualitative Results

We present further qualitative results in the following figures 11, 12, and 13. In Figure 11, the first two meshes are from the Real World Textured Things [21] dataset and the last mesh is generated from Hunyuan3D model [33]. This shows that our method is robust to simplify highly complex and non-manifold meshes. Similarly, in Figure 12 and 13, first mesh is from the Real World Textured Things [21] dataset and last two meshes are generated from Hunyuan3D model [33]. Note that wherever there is a cross, it signifies that this method is not able to simplify the given mesh beyond a certain number of faces.

In each diagram, we have provided first seven columns other than ground truth as our un-textured results to show our geometric accuracy in comparison to the other methods. Whereas the next three columns demonstrates the texture quality after simplification.

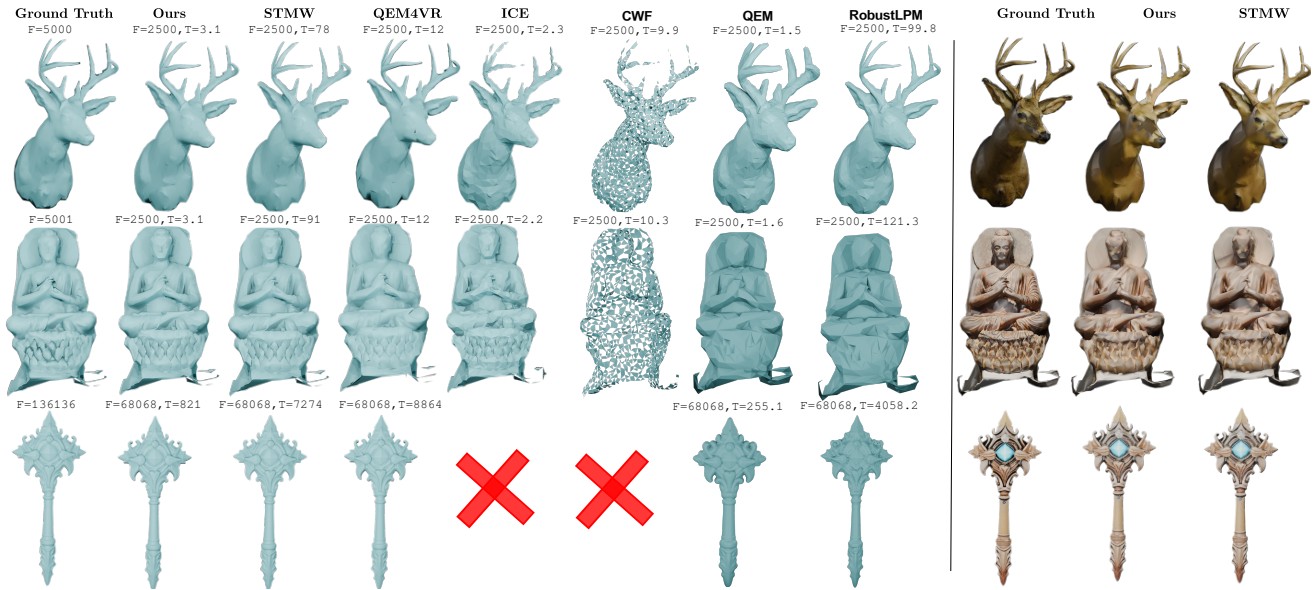

Figure 11. We present additional qualitative simplification results at 50% resolution of examples shown in our main paper Fig 4.

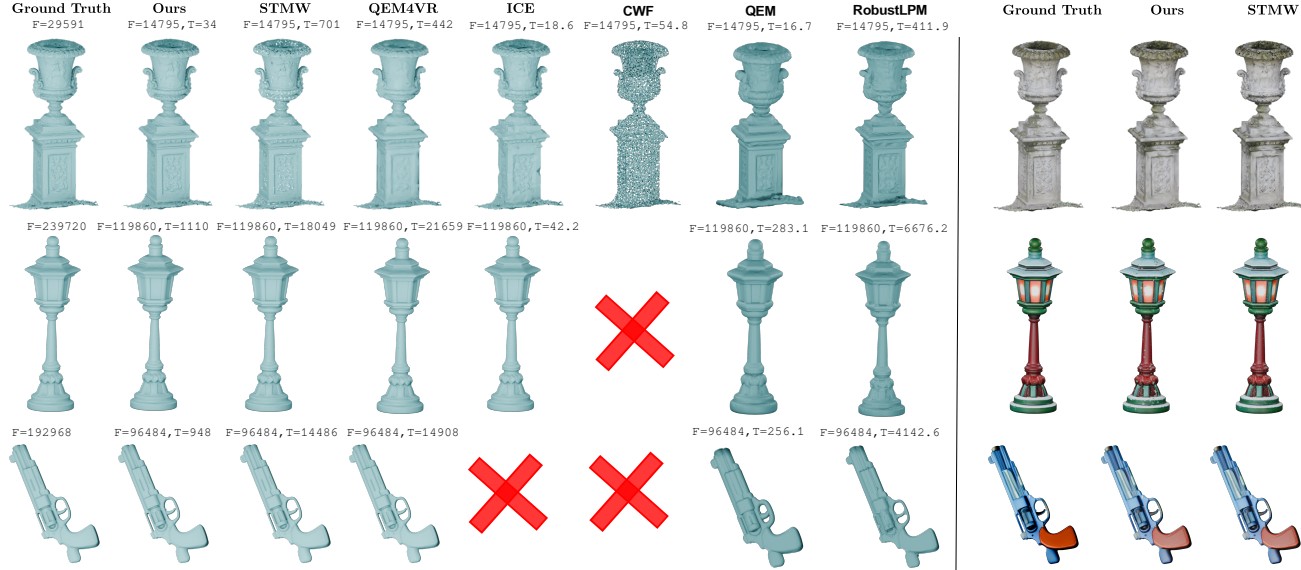

Figure 12. **Sample qualitative results at 50% resolution:** We present additional results 50% resolution. The first example is from Real-World Textured Things dataset [21], while the last two examples are AI generated using Hunyuan3D [33]. The results on 10% resolution of the these examples are shown in Figure 13.

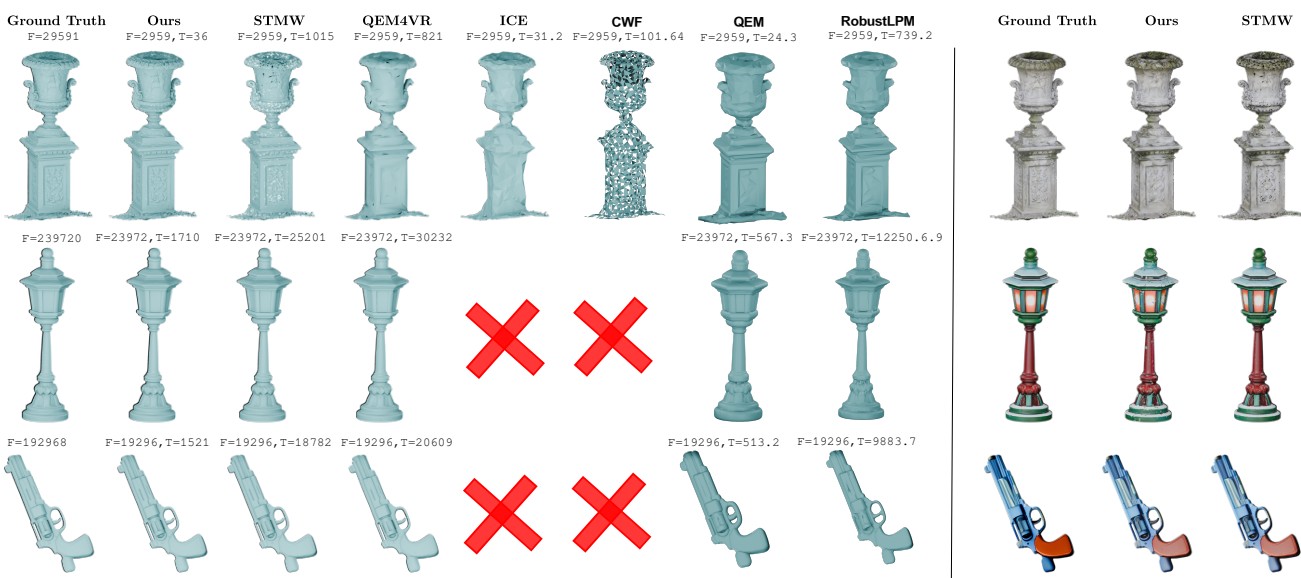

Figure 13. **Sample qualitative results at 10% resolution:** Results on 10% resolution of the examples shown in Figure 12.