# OpenReview forum: "Fast and Robust Mesh Simplification for Generated and Real-World 3D Assets"
_thecvf.com/CVPR/2026/Workshop/3D4S — CVPR 2026 Workshop 3D4S Oral_

### Official Review · Reviewer_7pc6 · 2026-04-13
**Training free and efficient way to optimize geometry but undermines texture**

**Rating:** 7
**Confidence:** 4

**Review:**

The authors propose a neat way to conserve meshes in the process of decimation by introducing additional constraints during optimization to address the following cases: 1. High frequency feature conservation, Boundary and Curvature Preservation, distortion reduction. It is an intuitive and well explained technique to preserve mesh shapes and features. They also consider and address the common failure scenarios like normal flipping or the need for virtual edges. The second stage of texture baking however is started off with sampling the surface and assigning point colors which is extremely lossy and if the end goal is decimation, then even more so. The authors should consider and evaluate any projection based baking techniques which can detach texture from geometry and so can have no texture degradation in the case of geometry change. Additionally the metric used to measure texture loss is point based which is heavily deficient.

---

### Official Review · Reviewer_RbHM · 2026-04-13
**A technically sound mesh simplification paper with strong speed and geometric fidelity results, though the gap framing and abstract claims need tightening**

**Rating:** 7
**Confidence:** 2

**Review:**

FA-QEM proposes a feature-aware extension of QEM for mesh simplification, combining boundary curvature preservation, normal consistency, and area weighting into a unified multi-term quadric formulation, paired with successive mapping for texture transfer. The method targets dense, noisy, and non-manifold meshes from reconstruction and generative pipelines (NeRF, 3DGS, etc.). The paper is technically sound and clearly written, with strong quantitative results on Thingi10K, 100% completion rate, and 3.5× faster than Liu et al., supported by a thorough ablation study and sensitivity analysis across six baselines. The individual components build on known techniques, but the unified composite quadric is a meaningful contribution, and the specific combination of speed and geometric fidelity on non-manifold inputs is genuine, even if the "trilemma" framing overstates novelty given partial overlap with CWF and SimpliGuard. The main concerns are that texture fidelity lags Liu et al. (0.099 vs. 0.078), which conflicts with the abstract's "better visual fidelity" claim, and that the abstract-to-results alignment could be tighter overall. The paper fits the workshop's stated topics on computational complexity, complex/unstructured geometries, and the efficiency-accuracy trade-off (Q3) well, though a scientific-domain experiment would further strengthen its relevance.

---

### Official Review · Reviewer_EyKT · 2026-04-23
**Review of FA-QEM**

**Rating:** 7
**Confidence:** 4

**Review:**

### Summary

The paper proposes FA-QEM, a feature-aware QEM pipeline for simplifying generated and real-world meshes. The method is practical and appears robust, but the conceptual novelty is modest.

### Strengths

- This problem is very practical and important. 3D generation and reconstruction do generate a large number of dense meshes, and simplification is a necessary step in any practical pipeline.
- The method is simple and achievable. It doesn't use complex learning methods and doesn't require training data. The core is QEM + priority queue + edge collapse.
- The experiments are relatively complete. It simultaneously examines geometry, texture, runtime, robustness, and ablation.

### Weaknesses

- The method is essentially an aggregation of existing QEM heuristics (curvature scaling, area preservation, normal penalties).
- The method relies on a fixed set of weights across all datasets. While the supplementary shows a relatively flat sensitivity curve, it is still unclear why these weights should generalize across meshes with different scales, noise levels, and geometric characteristics. Are the quadrics properly normalized to ensure scale invariance? How sensitive is the method to extreme cases (e.g., highly noisy scans and CAD models)?

This is a highly practical, systems/engineering-oriented paper. It doesn't present very deep new theories, but its objectives are clear, its methods are reasonable, its experimental coverage is good, and it has practical value for modern 3D generation pipelines.

---

### Official Review · Reviewer_GJH8 · 2026-04-24
**The paper proposes FA-QEM, an extended Quadric Error Metric pipeline that jointly optimizes for geometric deviation, boundary curvature, normal consistency, and area preservation. Coupled with a successive mapping strategy for texture transfer, the method achieves an impressive balance of speed, robustness, and visual fidelity on dense, noisy, and non-manifold meshes generated by modern 3D pipelines.**

**Rating:** 8
**Confidence:** 4

**Review:**

This paper addresses a highly relevant bottleneck in modern 3D vision and graphics: the simplification of dense, noisy, and non-manifold meshes produced by neural reconstruction (NeRF, 3DGS) and generative AI pipelines. The authors introduce Feature-Aware Quadric Error Metric (FA-QEM), which extends the classic QEM framework into a multi-objective optimization. The composite quadric $Q_{gf}$ encodes a base quadric (with inverse-area weighting to encourage simplification in flat regions), a curvature-scaled boundary quadric utilizing a dual-plane constraint, and a normal preservation quadric. To prevent boundary shrinkage, an independent area preservation cost is also utilized. Finally, the authors apply a successive mapping technique—tracing the lineage of edge collapses—to robustly transfer textures without the artifacts common to standard ray-projection methods.
Quality: The technical quality of the paper is high. The proposed composite quadric formulation is mathematically sound and effectively translates complex geometric preservation goals into efficient linear algebraic operations. The experimental validation is rigorous, leveraging the challenging Thingi10K dataset (for geometry and robustness) and the Real-World Textured Things dataset (for appearance). Achieving a 100% success rate on Thingi10K while maintaining polynomial runtime scaling ($O(N^2)$) is a significant engineering and algorithmic accomplishment.
Clarity: The paper is exceptionally well-written and easy to follow. The methodology clearly steps through each component of the composite cost function (Eq. 2-11). The distinction between the geometric feature quadric and the area preservation cost is well-motivated. The supplementary material excellently supports the main text with detailed sensitivity analyses and runtime profiling.
Originality: While the foundational concepts of QEM and successive mapping for texture transfer (adapted from Liu et al., STMW) are not new, the specific multi-term formulation is highly novel. In particular, the dual-plane boundary penalty scaled by discrete local curvature (Eq. 7-9) and the inverse-area weighting heuristic offer a fresh, highly effective take on feature-aware decimation.
Significance: With the recent explosion of 3D generative AI and radiance field extractions, there is a massive influx of "in-the-wild" assets that are entirely unsuitable for downstream tasks like simulation or gaming. A method that robustly simplifies these meshes while preserving textures—at speeds up to 14x faster than state-of-the-art robust baselines—is of immense practical value to the CVPR community and industry practitioners.
Pros

Exceptional Speed-Fidelity Trade-off: The method achieves state-of-the-art geometric error metrics (Hausdorff and Chamfer) while remaining drastically faster than competing robust methods like STMW and CWF.

High Robustness: Successfully processing 100% of the Thingi10K dataset down to 1% resolution proves the method's resilience against non-manifold geometry, duplicated edges, and noise. The virtual edge insertion strategy for merging disconnected components is a clever addition.

Excellent Texture Preservation: By utilizing successive mapping over an improved geometric "canvas," the method achieves highly competitive texture Chamfer distances without the computational overhead of direct appearance optimization.

Thorough Ablation: The ablation study (Table 4) and the supplementary sensitivity analysis comprehensively justify the inclusion of each term in the composite quadric.
Cons
Hyperparameter Dependency: The cost function relies on several weighting parameters ($w_{area}$, $w_{boundary}$, $w_{normal}$, $w_{plane\_area}$). While the authors demonstrate that the method is relatively insensitive to precise tuning and operates well with fixed defaults, highly pathological meshes outside the tested distributions might still require manual tweaking.Limited Novelty in Texture Transfer: The successive mapping technique is borrowed heavily from Liu et al. [17]. While its integration into this specific pipeline is effective, it limits the theoretical contribution of the second stage of the algorithm.Failure Cases on Interleaved Surfaces: As acknowledged by the authors, the virtual edge heuristic can fail or create ambiguous connections when dealing with extremely thin, self-intersecting, or interleaved surfaces (e.g., chain-link fences).

---

### Decision · Program_Chairs · 2026-04-28

Accept (Oral)